# SignAligner: Harmonizing Complementary Pose Modalities for Coherent Sign Language Generation

## Abstract

Sign language generation faces the challenge of producing natural and expressive results due to the complexity of sign language, which involves hand gestures, facial expressions, and body movements. In this work, we propose a novel method called SignAligner for realistic sign language generation. The framework consists of three stages: text-driven multimodal co-generation, online collaborative correction, and realistic video synthesis. First, a joint generator incorporating a Transformer-based text encoder and cross-modal attention simultaneously produces posture, gesture, and body movements from text. Next, an online correction module refines the generated modalities using dynamic loss weighting and cross-modal attention to resolve spatiotemporal conflicts and enhance semantic consistency. Finally, the corrected poses are input into a pre-trained video generation network to synthesize high-fidelity sign language videos. Additionally, we introduce a dataset extension scheme that derives three new landmark representations (*i.e.*, Pose, Hamer, and Smplerx) via pre-trained models, validated on PHOENIX14T and CSL-daily. Extensive experiments show that SignAligner significantly improves the accuracy and expressiveness of generated sign videos.

## 1 Introduction

Sign language is both a rich visual language and a primary form of communication within the deaf community. As a result, Sign Language Generation (SLG) is gaining significant attention in the field of visual languages and has become a classical yet challenging task. SLG encompasses various representational forms, including pose, avatar, and realistic video, each emphasizing different levels of action details and semantic representations.

Early SLG works primarily focus on avatar-based methods Baldassarri et al. (2009); Glauert et al. (2006), which require expensive pre-acquisition of poses due to rule-based lookups in a pre-set database. Given the critical role of pose in conveying the semantics of sign language, there has been a growing shift towards the study of text-to-pose generation Zelinka et al. (2019); Stoll et al. (2020); Krishna & Ukey (2021); Xiao et al. (2020). Inspired by the success of transformer models Unanue et al. (2021); Radford et al. (2021), Saunders *et al.* introduce a progressive transformer for end-to-end sign pose generation Saunders et al. (2020). Similarly, Huang *et al.* Huang et al. (2021) propose a non-autoregressive model with parallel decoding, alleviating the error accumulation and high inference latency issues of autoregressive models in previous G2P approaches. LVMCNWang et al. (2025) addresses the SLG challenge by bridging the modality semantic gap and addressing the lack of word-action correspondence labels. With the rapid advancements in diffusion models, Sign-IDD Tang et al. (2025a) leverages limb skeleton modeling to constrain joint associations and gesture details, significantly improving the accuracy and naturalness of generated poses.

With the booming development of large-scale pre-trained modeling techniques Fu et al. (2025); Li et al. (2022); Hu et al. (2023), research on generating sign language videos based on digital humans is becoming a cutting-edge hotspot. Saunders *et al.* Saunders et al. (2022) propose SIGNGAN, which directly maps parameterized skeletal pose sequences to high-fidelity sign language videos, achieving end-to-end integration of motion generation and rendering. Xie *et al.* Xie et al. (2024a) eliminate explicit pose representations and jointly train a video generator and a latent space decoder

Figure 1: Overview of the proposed SignAligner. It contains three stages: text-driven pose modalities co-generation, online collaborative correction of multimodality, and realistic sign video synthesis. First, a joint sign language generator produces three pose modalities: $\tilde{p}_{1:m}$, $\tilde{h}_{1:m}$, $\tilde{s}_{1:m}$, representing posture, handshape, and body motion. Next, an online collaborative correction mechanism refines these representations, enhancing their naturalness and spatial accuracy. Finally, a photo-realistic sign language video is synthesized using a pre-trained video synthesis network.

to directly generate realistic sign language videos. However, the existing research paradigms are still limited by the shackles of modal fragmentation: they are either confined to single-modal information representation or rely on multi-stage pipeline architectures, resulting in inherent shortcomings such as reduced semantic fidelity and lack of multimodal co-evolutionary mechanisms in the sign language generation process. Specifically, the separate processing of gestures and avatars ignores the multimodal coupling of sign language, while the staged generation strategy weakens motion details due to quantization loss of intermediate representations, undermining the spatiotemporal continuity that sign language is supposed to maintain. As shown in Table 1, real sign language videos better reflect its essential semantic properties than any single-modal **Pose**, **Hamer** and **Smplerx**. Therefore, building a unified sign language video generation model is crucial.

To this end, we propose a novel method termed *SignAligner* for realistic sign language generation. As shown in Figure 1, SignAligner consists of three stages: text-driven pose modalities co-generation, online collaborative correction, and realistic sign video synthesis. Firstly, by combining text information, three sign language representation generators are used to simultaneously obtain posture coordinates, gesture actions, and body motion trajectories. Here, the text encoder uses Transformer architecture to extract semantic features, and the generation module combines text semantic information through cross modal attention mechanism, and simultaneously generates multi-source sign language representations to ensure accurate mapping and diversity control of modal features. In online collaborative correction of multimodality, we use a cross-modal attention mechanism and a dynamic loss weighting strategy to optimize the generated pose modalities, achieving information complementarity between different modalities, dynamically eliminating spatiotemporal conflicts between modalities, and ensuring semantic coherence and action consistency of the generated results. Finally, the corrected pose modalities are input into a pre-trained synthesis network to obtain high-fidelity sign videos. Our main contributions are summarized as follows:

- To address the limited and homogeneous nature of existing sign language datasets, we propose a dataset expansion scheme that integrates multiple sign language representations through pre-trained models, expanding widely used sign language corpora. These include high-precision skeleton data with facial keypoints (**Pose**), hand details (**Hamer**), and 3D full-body posture (**Smplerx**), which aims to provide a novel, diverse, and high-quality resource for the sign language community.

- We establish a three-stage sign video generation method, which utilizes the complementarity among pose modalities and introduces textual information to guide the generation process, thereby achieving sign language videos that can more accurately reflect the semantic content of the text.

- We propose a joint generation mechanism that collaboratively generates sign language representations and utilizes a multimodal correction strategy combined with dynamic loss constraints to fully explore complementary information between modalities, ensuring higher quality sign language video generation in terms of semantic consistency, naturalness of actions, and spatial expression accuracy.

## 2 RELATED WORK

### 2.1 SINGLE-STAGE SIGN LANGUAGE GENERATION

Sign language is both a rich visual language and a preferred mode of communication for the deaf community. With the growing need for effective communication between deaf and hearing people in recent decades, Sign Language Gener-

Table 1: Semantic evaluation performance of different sign language representations on the PHOENIX14T dataset.

| Ground Truth | BLEU-1↑ | BLEU-2↑ | BLEU-3↑ | BLEU-4↑ | ROUGE↑ |
|---|---|---|---|---|---|
| Pose | 30.11 | 20.86 | 15.70 | 12.50 | 29.68 |
| Hamer | 31.90 | 22.43 | 16.68 | 13.24 | 31.21 |
| Smplerx | 30.18 | 20.40 | 14.92 | 11.84 | 28.91 |
| Video | 38.45 | 28.23 | 21.65 | 17.42 | 37.65 |

ation (SLG) Cui et al. (2019); Krishna & Ukey (2021); Stoll et al. (2020); Krishna & Ukey (2021); Saunders et al. (2020); Tang et al. (2025b) have received significant attention in recent years. Current work focuses on single-stage SLG, *i.e.*, text-to-pose, text-to-avatar, and text-to-video.

**Text to Pose.** Sign language pose videos are widely used because they can capture the semantic and dynamic characteristics of actions through sequences of skeletal key points. Inspired by the great success of transformer Unanue et al. (2021); Radford et al. (2021); Yang et al. (2023a), Saunders *et al.* design a progressive transformer to generate sign poses in an end-to-end manner Saunders et al. (2020). LVMCN Wang et al. (2025) solves the semantic gap between modalities and the lack of word-action correspondence labels required for strong supervised alignment in SLG. With the rapid development of diffusion models, Xie *et al.* Xie et al. (2024b) ingeniously combine VAE with vector quantization to propose Pose-VQVAE, which effectively generates discrete potential representations for continuous pose sequences. Sign-IDD Tang et al. (2025a) enhances the accuracy and naturalness of pose generation by constraining joint associations and gesture details through skeletal modeling.

**Text to Avatar.** Early SLG studies often rely on avatar-based approaches, as a single skeletal representation is insufficient for realistic visual presentation. Baldassarri *et al.* Baldassarri et al. (2009) develop an animation engine that enables avatars to adapt signs and expressions to the interpreter's mood. Glauert *et al.* Glauert et al. (2006) design the VANESSA system, which converts speech or text into virtual sign language avatars. T2S-GPT Yin et al. (2024) introduces a dynamic vector quantization DVA-VAE that adjusts encoding length to the density of sign information, generating corresponding 3D avatars. Baltatzis *et al.* Baltatzis et al. (2024) further advance this direction by combining SMPL-X with a graph neural network in a diffusion model, producing dynamic 3D avatar sequences from unconstrained inputs and pushing SLG closer to realistic neural avatars.

**Text to Video.** Thanks to the rapid development of artificial intelligence technology, generating realistic sign language videos has gradually become possible. Kaur *et al.* Kaur & Kumar (2016) develop HamNoSys, a sign language transcription system that can be directly mapped to an avatar, and each of its symbols contains a description of the initial posture and the movement over time. Xie *et al.* Xie et al. (2024a) develop a new method to produce high-quality sign language videos without the intermediate step of human pose. It first learns from the generator and the hidden features of the video, and then uses another model to understand the order of these hidden features.

### 2.2 MULTI-STAGE SIGN LANGUAGE GENERATION

The first deep learning-based SLG pipeline decomposes the task into three stages: Text-to-Gloss (T2G) translation, Gloss-to-Pose (G2P) generation, and Pose-to-Video (P2V) synthesis Stoll et al. (2020). Building on this paradigm, Brock *et al.* Brock et al. (2020) generate 3D kinematic skeletons from monocular video and estimate joint angular displacements via inverse kinematics to animate virtual sign characters. Stoll *et al.* Stoll et al. (2018) develop a system that translates speech into gloss sequences with an encoder–decoder network, maps gloss to pose through data-driven learning, and synthesizes sign language videos driven by the generated poses. Saunders *et al.* Saunders et al. (2022) further advance this line with FS-NET and SIGNGAN, enabling direct video generation from skeletal inputs. Despite these advances, existing methods remain constrained by reliance on single modalities or limited information representations, making it difficult to capture the full semantic and visual richness of sign language. To address this, we propose a multi-stage SLG framework that leverages text to guide the generation of multimodal representations, ultimately producing videos with greater realism and semantic consistency.

## 3    DATASET CURATION

Since the variety of datasets available for existing sign language generation tasks is relatively limited and monolithic in form, most of them are confined to only videos and the corresponding skeleton coordinates. Not only that, with the rapid development of large-scale, more and more demands pay more attention to the generation of real-life sign language videos, but only the skeleton form of the data can not do such an effect, lack of many details, can not meet the needs of reality. Therefore, we formally augment the widely used sign language datasets, German sign language corpus PHOENIX14T Camgoz et al. (2018) and Chinese Sign Language Corpus CSL-daily Zhou et al. (2021), using pre-trained models DWPose Yang et al. (2023b), HaMeR Pavlakos et al. (2024) and SMPLer-X Cai et al. (2023), respectively. Specifically, this includes high-precision skeleton data (**Pose**) containing facial keypoints, 3D full-body pose (**Smplerx**), and (**Hamer**) reflecting hand details, aiming to provide the community with a novel and diverse large-scale sign language corpus suitable for both practical applications and academic research. Figure 2 shows the three types of sign representations from the PHOENIX14T dataset. For more details, please refer to the section A.2.4.

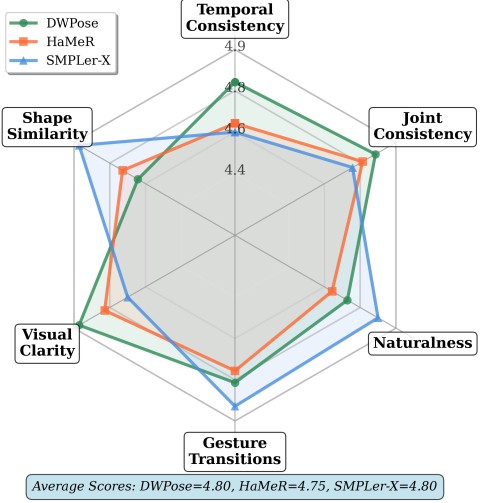

Figure 2: Examples of the PHOENIX14T.

**Extraction quality discussion.** We systematically discuss the quality of data extraction from two perspectives: (1) objective performance analysis based on experimental details reported in the original paper; and (2) correspondence analysis based on subjective perception scores from user studies. On the one hand, for the three models DWPose, HaMeR, and SMPLer-X, we observe that they are all fully trained and tested on large-scale and diverse datasets, such as 3DPW Von Marcard et al. (2018), MTC Xiang et al. (2019), FreiHAND Zimmermann et al. (2019), COCO Jin et al. (2020), HO3D Hampali et al. (2020), InterHand2.6M Moon et al. (2020), AGORA Patel et al. (2021), Halpe Fang et al. (2022) and UBody Lin et al. (2023). These datasets cover a wide range of scenarios, from everyday natural scenes to specialized tasks such as hand manipulation and multi-person interaction. They include both high-resolution 2D image annotations and detailed 3D pose and mesh annotations, with a sample size

Figure 3: User study on extraction quality.

of millions, ensuring the model's generalization and robustness in complex environments. Based on this comprehensive data foundation, three models achieve superior results across multiple metrics in a systematic comparison with dozens of mainstream methods, demonstrating the advanced quality of our methods across diverse data extraction capabilities. On the other hand, we also recruit 100 volunteers with varying levels of sign language proficiency to further validate the quality of keypoint extraction from a subjective perspective. Specifically, we use six key evaluation dimensions: Temporal Consistency, Shape Similarity, Visual Clarity, Gesture Transitions, Naturalness, Joint Consistency. Each volunteer independently scores 10 random samples from the three modalities (out of 5 points) to fully reflect the perceptual differences between different audiences during actual viewing. As shown in the Figure 3, our extracted modalities consistently achieved high subjective scores exceeding 4.0, demonstrating their superior visual presentation and dynamic coherence. This also confirms the effectiveness of our objective performance evaluation from a perceptual perspective.

## 4 METHOD

This work introduces SignAligner, a novel framework for generating realistic sign language videos from text. It consists of three stages: text-driven pose modalities co-generation, online collaborative correction and realistic sign video synthesis. A joint sign generator integrates textual semantics via a Transformer encoder and cross-modal attention to produce pose, hamer and smplerx. Then, an online correction strategy weighted by a dynamic loss enforces inter-modality complementarity. High-fidelity sign language videos are then rendered using a pre-trained video generator.

### 4.1 TEXT-DRIVEN POSE MODALITIES CO-GENERATION.

**Text Semantic Feature Extraction.** To extract the semantic features of text, we build a transformer-based encoder. A linear embedding layer is adopted to map text into a high-dimensional feature space. We further apply a positional coding layer to complement the temporal order of the text vectors. The computation is as follows:

$$t'_n = W^t \cdot t_n + b^t + \text{PE(n)}, \tag{1}$$

where $t_n$ is a one-hot vector of the n-th text over the text vocabulary $\mathcal{V}$, $PE$ is conducted by the sine and cosine functions on the temporal text and pose order as in Vaswani et al. (2017). $W^t$ and $b^t$ represent the weight and bias respectively.

Next, we input the obtained text embeddings $\{t'\}_{n=1}^N$ into the TextEncoder to capture the global semantics of the text. Here, the encoder consists of n identical blocks which include Multi-Head Attention, Normalisation and Feedforward Layers. The calculation process can be expressed as:

$$\tilde{t}_{1:N} = TextEncoder(t'_{1:N}). \tag{2}$$

The self-attention mechanism computes contextual dependencies using scaled dot-product attention:

$$\text{Attention}(Q, K, V) = \text{softmax}\left(\frac{QK^\top}{\sqrt{d}}\right) V, \tag{3}$$

where $Q, K, V \in \mathbb{R}^{d \times d}$ are the query, key, and value matrices derived from $\mathbf{t}'_n$, and $d$ is the feature dimension.

**Pose Modalities Co-generation.** In this work, we aim to simultaneously generate pose modalities including **Pose**, **Hamer**, and **Smplerx** for temporal consistency. Therefore, similar to text encoding, we encode pose, hamer and smplerx as high-dimensional feature spaces through linear and positional encoding layers $\text{PE}_p$, $\text{PE}_h$ and $\text{PE}_s$, respectively. The computational formula is as follows:

$$p'_m = W^p \cdot p_m + b^p + \text{PE}_p(m); \ h'_m = W^h \cdot h_m + b^h + \text{PE}_h(m); \ s'_m = W^s \cdot s_m + b^s + \text{PE}_s(m), \tag{4}$$

where $p_m$, $h_m$ and $s_m$ denote the coordinates of pose, hamer and smplerx at m-th timestamp, respectively. $W^p$, $W^h$, $W^s$ and $b^p$, $b^h$, $b^s$ denote the learnable weights and bias, respectively.

To avoid spatiotemporal discrepancies between modalities and generate three sign language representations simultaneously, we input visual features into a multimodal joint training framework built on Transformer. This framework consists of three components: PoseDecoder, HamerDecoder, and SmplerxDecoder. Take the PoseDecoder as an example, it can predict the next pose $\tilde{p}_{m+1}$ by aggregating all previously generated poses $\tilde{p}_{1:m}$.

We also use an cross-attention mechanism in the transformer to enable semantic interaction between textual and visual sequences. The various decoders can be represented as:

$$\tilde{p}_{m+1} = \text{PD}(p'_{1:m}, \tilde{t}_{1:N}); \ \tilde{h}_{m+1} = \text{HD}(h'_{1:m}, \tilde{t}_{1:N}); \ \tilde{s}_{m+1} = \text{SD}(s'_{1:m}, \tilde{t}_{1:N}), \tag{5}$$

where PD, HD and SD stand for PoseDecoder, HamerDecoder and SmplerxDecoder respectively.

After $M$ time stamps, we obtain the pose representation $\{\tilde{p}\}_{m=1}^M$, hamer representation $\{\tilde{h}\}_{m=1}^M$, smplerx representation $\{\tilde{s}\}_{m=1}^M$. In the training stage, the Mean Absolute Error (MAE) loss is used to constraint the consistency of the generated poses $\{\tilde{p}\}_{m=1}^M$, generated hamers $\{\tilde{h}\}_{m=1}^M$ and generated smplerxs $\{\tilde{s}\}_{m=1}^M$, respectively, with the ground truth $\widehat{P} = \{\widehat{p}\}_{m=1}^M$, $\widehat{H} = \{\widehat{h}\}_{m=1}^M$ and $\widehat{S} = \{\widehat{s}\}_{m=1}^M$.

$$\mathcal{L}_{\text{TMC}} = \frac{1}{M} \sum_{m=1}^M (|\tilde{p}_m - \widehat{p}_m| + |\tilde{h}_m - \widehat{h}_m| + |\tilde{s}_m - \widehat{s}_m|). \tag{6}$$

## 4.2 ONLINE COLLABORATIVE CORRECTION

Direct fusion of heterogeneous modalities introduces inconsistency and noise, while separate decoders limit feature learning. To address this, we propose an online multimodal correction strategy that dynamically regulates inter-modal correlations, enhancing robustness and interaction. A triple cross-modal attention pathway exploits skeleton priors to refine hand features and uses local semantics to enhance full-body spatio-temporal representations. Dynamic complementarity is achieved by back-optimizing skeleton confidence with global pose information, as formulated below:

$$\tilde{p}'_{1:m} = \text{CA}(\tilde{p}_{1:m}, \tilde{h}_{1:m}, \tilde{s}_{1:m}); \tilde{h}'_{1:m} = \text{CA}(\tilde{h}_{1:m}, \tilde{p}_{1:m}, \tilde{s}_{1:m}); \tilde{s}'_{1:m} = \text{CA}(\tilde{s}_{1:m}, \tilde{p}_{1:m}, \tilde{h}_{1:m}), \tag{7}$$

where CA stand for cross-attention.

In the correction stage, we design a dynamic loss weighting module that adaptively adjusts the importance of each loss. Specifically, learnable parameters $\alpha, \beta, \gamma \in \mathbb{R}^+$ are normalized via softmax to obtain weight coefficients, and are automatically updated through backpropagation to achieve dynamic synergy and adaptive optimization across modalities:

$$w_A, w_B, w_C = softmax(\alpha, \beta, \gamma) = [\frac{e^\alpha}{e^\alpha + e^\beta + e^\gamma}, \frac{e^\beta}{e^\alpha + e^\beta + e^\gamma}, \frac{e^\gamma}{e^\alpha + e^\beta + e^\gamma}], \tag{8}$$

where $w_A$, $w_B$ and $w_C$ are the weights of modes $Pose$, $Hamer$ and $Smplerx$ respectively.

Finally, to enforce semantic consistency and cross-modal complementarity, we apply adaptive weights to dynamically constrain the generative features of the three modalities, ensuring that the refined sign language representations align more closely with real semantics.

$$\mathcal{L}_{\text{OMC}} = w_A \cdot ||\tilde{p}'_m - \hat{p}_m||_2^2 + w_B \cdot ||\tilde{h}'_m - \hat{h}_m||_2^2 + w_C \cdot ||\tilde{s}'_m - \hat{s}_m||_2^2, \tag{9}$$

where $\tilde{p}'^M_{m=1}$, $\tilde{h}'^M_{m=1}$ and $\tilde{s}'^M_{m=1}$ are generated features after calibration, and $\hat{p}^M_{m=1}$, $\hat{h}^M_{m=1}$ and $\hat{s}^M_{m=1}$ are the corresponding real labels.

## 4.3 REALISTIC SIGN VIDEO SYNTHESIS

In order to generate highly realistic sign language videos, we adopt the RealisDance Zhou et al. (2024) and retrain it with PHOENIX14T and CSL-daily to meet the specific needs of sign language video generation. First, we pass the corrected pose, hamer and smplerx through a gating module, and integrate this information to fine-tune the RealisDance main framework. The specific process can be expressed as follows:

$$Video = RealisDance(\tilde{p}'_{1:m}, \tilde{h}'_{1:m}, \tilde{s}'_{1:m}). \tag{10}$$

In the model training process, we define a joint loss function $L_{\text{EVS}}$, consisting of $L_{\text{rec}}$ and loss $L_{\text{adv}}$.

$$L_{\text{EVS}} = L_{\text{rec}} + \lambda L_{\text{adv}}, \tag{11}$$

where the reconstruction loss $L_{\text{rec}}$ measures the difference between the generated video and the real video, the adversarial loss $L_{\text{adv}}$ improves the realism of the generated video by introducing a discriminator, and $\lambda$ is a weighting coefficient to balance the two.

## 5 EXPERIMENTS

### 5.1 EXPERIMENTAL SETTINGS

**Implementation Details.** In the text-driven pose modalities co-generation, we use a Transformer-based generative model with the Adam optimizer for training. For the OCC, the cross-modal multi-head attention mechanism is configured with 2 layers, 4 attention heads batch size of 64. Finally, end-to-end video generation is achieved using the RealisDance Zhou et al. (2024), retrained on the PHOENIX14T and CSL-daily datasets. All experiments are conducted on 8 NVIDIA A40 GPUs.

**Evaluation Metrics.** We adopt NSLT Camgoz et al. (2018) as an offline back-translation tool for pose evaluation, following prior work Saunders et al. (2020); Tang et al. (2022). Since Hamer, Smplerx, and Video rely on video modality rather than pose keypoints, we employ GFSLT Zhou et al.

Table 2: Quantitative results on the PHOENIX14T dataset.

| Methods | DEV | | | | | | TEST | | | | | |
|---|---|---|---|---|---|---|---|---|---|---|---|---|
| | BLEU-1↑ | BLEU-4↑ | ROUGE↑ | SSIM↑ | PSNR↑ | FID↓ | BLEU-1↑ | BLEU-4↑ | ROUGE↑ | SSIM↑ | PSNR↑ | FID↓ |
| PTSLP Saunders et al. (2020) | 8.55 | 1.68 | 9.15 | 0.584 | 11.282 | 51.643 | 8.86 | 1.52 | 8.83 | 0.584 | 11.452 | 52.122 |
| GEN-OBT Tang et al. (2022) | 13.70 | 5.47 | 16.08 | 0.690 | 13.772 | 30.114 | 13.31 | 4.94 | 14.32 | 0.689 | 13.842 | 32.231 |
| CogvideoX Yang et al. (2024) | 8.14 | 0.46 | 7.21 | 0.292 | 3.822 | 263.823 | 8.40 | 0.51 | 7.33 | 0.287 | 3.819 | 264.751 |
| LVMCN Wang et al. (2025) | 12.61 | 4.94 | 14.34 | 0.689 | 13.726 | 31.667 | 14.57 | 5.61 | 16.07 | 0.689 | 13.853 | 34.278 |
| SignAligner (Ours) | **19.33** | **7.36** | **21.08** | **0.729** | **15.292** | **25.978** | **20.56** | **8.17** | **20.88** | **0.731** | **15.322** | **26.257** |

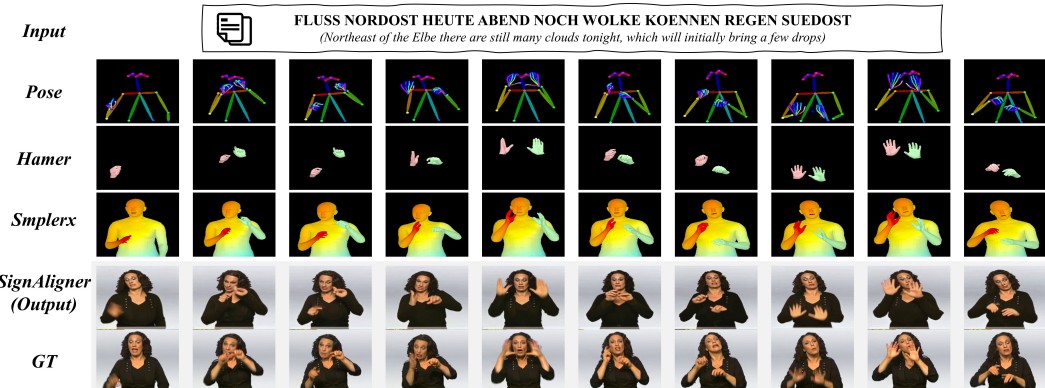

Figure 4: Visualization examples of produced sign language video sequence of SignAligner.

(2023) and retrain it on PHOENIX14T and CSL-Daily for direct video evaluation. To comprehensively assess our method, we combine semantic and visual quality metrics. At the semantics level, we report **BLEU** Papineni et al. (2002), **ROUGE** Lin (2004), and **WER** Wang et al. (2025), where BLEU measures semantic completeness and fluency via *n*-gram recall. At the vision level, we adopt **SSIM** Wang et al. (2004), **PSNR** Hore & Ziou (2010), and **FID** Heusel et al. (2017) to quantify visual similarity, detail fidelity, and distribution alignment with real videos.

## 5.2 COMPARISON WITH STATE-OF-THE-ARTS

**Comparison on PHOENIX14T.** As shown in Table 2, our multi-stage SLG method consistently outperforms existing approaches on PHOENIX14T. Compared with baseline PTSLP, SignAligner achieves substantial gains across all metrics, with BLEU-1 and ROUGE improvements of 10.78% and 11.83% on the DEV/TEST sets. We still achieve considerable improvement over the state-of-the-art sign language generation method, LVMCN, and a fine-tuned CogvideoX. Beyond language accuracy, SignAligner also enhances video quality, achieving an SSIM of 0.731 and PSNR of 15.322 on the test set, surpassing PTSLP in visual fidelity and clarity. As shown in Figure 4, this highlights SignAligner's robustness in generating semantically accurate and high-quality sign language videos.

We further provide a visualization example to clearly show the performance differences of different methods in sign video generation, as shown in Figure 5. Compared to PTSLP and CogvideoX, SignAligner demonstrates significant advantages, particularly in hand structure, movement trajectory, and spatiotemporal coordination. PTSLP and CogvideoX often suffer from blurred hands, misaligned postures, and discontinuous movements, making fine gesture details and movement continuity difficult to restore. In contrast, SignAligner produces more natural hand movements and smoother transitions, closely matching the ground truth. This convincingly demonstrates SignAligner's effectiveness in improving clarity, accuracy, and consistency in generated sign language videos through multimodal collaborative modeling and realistic video synthesis.

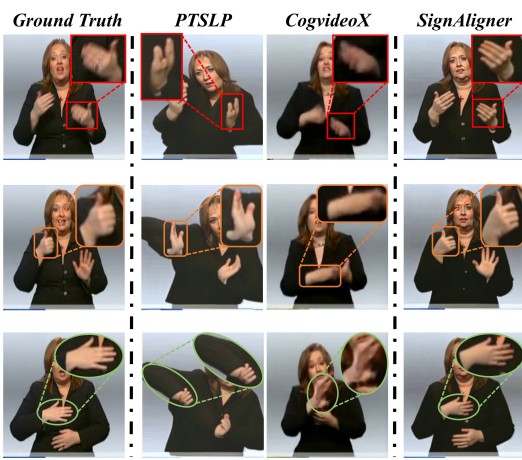

Figure 5: Visualization examples of PTSLP, CogvideoX and SignAligner on PHOENIX14T.

**Comparison on CSL-Daily.** As shown in Table 3, SignAligner achieves superior performance on CSL-daily. It obtains SSIM of 0.839, PSNR of 14.753, and FID of 26.589. Compared with existing methods, gains in SSIM and PSNR highlight its ability to capture structural features and ensure reconstruction quality, while the lower FID confirms the authenticity and diversity of generated videos, demonstrating overall superiority in sign language generation.

Table 3: Quantitative results on the CSL-daily.

| Methods | SSIM↑ | PSNR↑ | FID↓ |
|---|---|---|---|
| PTSLP Saunders et al. (2020) | 0.817 | 14.073 | 36.575 |
| GEN-OBT Tang et al. (2022) | 0.823 | 14.072 | 33.288 |
| LVMCN Wang et al. (2025) | 0.826 | 14.402 | 33.584 |
| SignAligner (Ours) | **0.839** | **14.753** | **26.589** |

Furthermore, as shown in Figure 6, we also conducted a qualitative comparison on the CSL-daily dataset. The results clearly demonstrate that our proposed SignAligner generates significantly more accurate, natural, and visually coherent gestures. Specifically, PTSLP frequently produces blurry or distorted gestures, while LVMCN often suffers from severe spatial misalignment, particularly in fast or complex motion situations. In particular, the hands in the third row of PTSLP's results are almost completely misaligned and visually implausible. In contrast, SignAligner better preserves the structural integrity of both hands and achieves higher temporal consistency. This improvement is evident from the more accurate and fine-grained finger details observed in the zoomed-in area, thereby further validating the robustness and effectiveness of our approach.

Figure 6: Visualization examples of PTSLP, LVMCN and SignAligner on CSL-daily.

**User Study.** We further conducted subjective evaluations on the PHOENIX14T and CSL-Daily datasets, focusing on four key metrics: naturalness, visual clarity, temporal consistency, and smoothness.

Table 4: User study comparison.

| Methods | Temporal consistency | Visual clarity | Gesture transitions | Naturalness |
|---|---|---|---|---|
| PTSLP | 2.09 | 1.92 | 2.03 | 2.17 |
| GEN-OBT | 3.59 | 3.34 | 3.66 | 3.61 |
| LVMCN | 3.32 | 3.28 | 3.51 | 3.45 |
| CogvideoX | 1.36 | 2.61 | 2.45 | 1.99 |
| SignAligner (Ours) | **4.27 (19%↑)** | **4.11 (23%↑)** | **4.23 (16%↑)** | **4.36 (21%↑)** |

In the experiments, 100 volunteers with varying levels of sign language proficiency rated the generation results of the five comparison methods. Each evaluation set included anonymous, randomly sorted samples from PTSLP, GEN-OBT, LVMCN, CogVideoX, and our proposed SignAligner, along with corresponding ground-truth reference videos as a control baseline. Scoring uses a scale of 1 to 5 to ensure quantitative evaluation of different dimensions. Table 4 shows that SignAligner outperforms other methods across all four dimensions, particularly achieving a significant improvement of up to 23% in visual clarity, fully demonstrating its superiority in perceptual quality.

## 5.3 ABLATION STUDY

**The impact of co-gen and OCC.** To verify the effectiveness of the proposed pose modalities co-generation (**co-gen**) and Online Collaborative Correction (**OCC**) mechanisms, we conduct ablation experiments on the

Table 5: Ablation results on PHOENIX14T dataset.

| Methods | BLEU-1↑ | BLEU-4↑ | ROUGE↑ | SSIM↑ | PSNR↑ | FID↓ |
|---|---|---|---|---|---|---|
| w/o co-gen | 14.50 | 5.11 | 15.54 | 0.675 | 13.791 | 38.158 |
| w/o OCC | 17.84 | 7.07 | 19.01 | 0.698 | 14.673 | 37.229 |
| SignAligner (Ours) | **20.56** | **8.17** | **20.88** | **0.731** | **15.322** | **26.257** |

PHOENIX14T dataset. As shown in Table 5, removing (**co-gen**) reduces BLEU-1 to 14.50% and BLEU-4 to 5.11%, while SSIM and PSNR also drop notably, confirming its role in enhancing language accuracy and visual fidelity. Without it, the model struggles to integrate posture information, leading to reduced coherence and accuracy. Similarly, removing (**OCC**) lowers BLEU-1 to 17.84% and ROUGE to 19.01%, together with declines in visual quality, highlighting the importance of real-time correction for semantic consistency. The full model achieves the best results across all metrics, validating the synergistic effect of co-generation and online collaborative correction.

**Generate more accurate skeleton pose.**
Here, as shown in Table 6, our proposed
SignAligner achieves significantly better per-
formance than existing models in the Text
to Pose task, reaching BLEU-1 24.39% and
ROUGE 25.21%. This shows that our method
can not only effectively capture the semantic
information in the text, but also generate ac-
tion sequences with good language structure
and coherence. The high semantic fidelity and sequence generation quality fully verify our design
advantages in multimodal modeling and cross-modal alignment, and further prove the strong poten-
tial of SignAligner in text-driven sign language generation tasks.

Table 6: Results on PHOENIX14T for Text to Pose.

| Methods | BLEU-1↑ | BLEU-4↑ | ROUGE↑ | WER↓ |
|---|---|---|---|---|
| PTSLP Saunders et al. (2020) | 13.35 | 4.31 | 13.17 | 96.50 |
| NAT-AT Huang et al. (2021) | 14.26 | 5.53 | 18.72 | 88.15 |
| NAT-EA Huang et al. (2021) | 15.12 | 6.66 | 19.43 | 82.01 |
| DET Viegas et al. (2023) | 17.18 | 5.76 | 17.64 | – |
| G2P-DDM Xie et al. (2024b) | 16.11 | 7.50 | – | 77.26 |
| GCDM Tang et al. (2025b) | 22.03 | 7.91 | 23.20 | 81.94 |
| GEN-OBT Tang et al. (2022) | 23.08 | 8.01 | 23.49 | 81.78 |
| SignAligner (Ours) | **24.39** | **8.47** | **25.21** | **73.89** |

Table 7: Results on PHOENIX14T dataset for Text to Hamer/Smplerx tasks.

| Methods | Text to Hamer | | | | | Text to Smplerx | | | | |
|---|---|---|---|---|---|---|---|---|---|---|
| | BLEU-1↑ | ROUGE↑ | SSIM↑ | PSNR↑ | FID↓ | BLEU-1↑ | ROUGE↑ | SSIM↑ | PSNR↑ | FID↓ |
| PTSLP Saunders et al. (2020) | 13.26 | 13.03 | 0.948 | 19.061 | 25.652 | 9.89 | 9.65 | 0.792 | 16.228 | 7.584 |
| GEN-OBT Tang et al. (2022) | 22.44 | 21.87 | 0.951 | 19.769 | 19.999 | 25.87 | 25.43 | 0.803 | 16.844 | 3.968 |
| CogvideoX Yang et al. (2024) | 15.16 | 14.17 | 0.915 | 14.932 | 36.587 | 9.85 | 7.67 | 0.724 | 12.033 | 43.228 |
| LVMCN Wang et al. (2025) | 22.20 | 22.23 | 0.951 | 19.903 | 22.475 | 23.75 | 24.25 | 0.807 | 17.047 | 4.106 |
| SignAligner (Ours) | **29.94** | **29.12** | **0.958** | **21.314** | **4.428** | **27.48** | **27.43** | **0.832** | **18.651** | **3.607** |

**Capture fine finger details.** As shown in Table 7, SignAligner achieves the best performance
across all metrics on the Text to Hamer task. BLEU-1 reaches 29.94%, markedly higher than the
baseline method PTSLP with 13.26%, and it significantly outperforms the widely used text-driven
video model CogVideoX. Moreover, SignAligner demonstrates superior video generation quality,
achieving an SSIM of 0.958, a PSNR of 21.314, and reducing FID to 4.428, substantially better than
CogVideoX, which had an FID of 36.587.

**Maintain stronger body expressiveness.** We further evaluated the model on the Text to Smplerx
task. As shown in Table 7, our model achieved the best performance. It excelled in video quality
metrics, achieving an SSIM of 0.832, a PSNR of 18.651, and an FID of 3.607.

**The importance of trimodal com-
plementarity.** To address the spe-
cific contributions of each modality
in the multimodal architecture: by re-
moving the pose, hamer, and smplerx
modalities separately, we evaluated
the final performance of the model under the same experimental setup. As shown in Table 8, no
single modality significantly degrades the quality of the generated videos. Through the generated
videos, we further discovered that when pose is missing, facial orientation is misaligned and the
face appears backward; when hamer is missing, fingers are undermodeled and multiple joints are
distorted; and when smplerx is missing, body shape is deviated and becomes unnatural. These
phenomena confirm the irreplaceable role of each modality in the framework: pose provides essen-
tial spatial localization for sign language movements, hamer ensures the semantic accuracy of fine
hand movements, and smplerx maintains the natural coordination of full-body posture. The three
complement each other, supporting a complete mapping of multimodal information to video.

Table 8: Results for missing modalities on PHOENIX14T.

| Methods | BLEU-1↑ | BLEU-4↑ | ROUGE↑ | SSIM↑ | PSNR↑ | FID↓ |
|---|---|---|---|---|---|---|
| w/o Pose | 14.35 | 4.19 | 6.63 | 0.637 | 13.285 | 50.234 |
| w/o Hamer | 11.70 | 4.17 | 13.66 | 0.682 | 13.848 | 45.071 |
| w/o Smplerx | 17.65 | 6.52 | 16.33 | 0.704 | 14.129 | 41.245 |
| SignAligner (Ours) | **20.56** | **8.17** | **20.88** | **0.731** | **15.322** | **26.257** |

# 6 CONCLUSIONS

In this work, we proposed a dataset extension scheme and expanded the PHOENIX14T and CSL-
daily datasets to include three gesture representations: Pose, Hamer, and Smplerx, aiming to improve
the expressiveness and realism of sign language generation. Building on this, we propose a novel
method called SignAligner for realistic sign language generation. By leveraging a Transformer-
based text encoder and a cross-modal attention mechanism, SignAligner effectively models semantic
and motion relationships across pose modalities. The online collaborative correction further refines
consistency and realism through dynamic loss adjustment. Finally, these refined representations
are synthesized into accurate and realistic sign language videos. Experimental results show that
SignAligner significantly outperforms existing methods in both accuracy and expressiveness.

## ETHICS STATEMENT

This work does not involve human subjects, animal experiments, or sensitive personal data. The datasets used (*e.g.*, PHOENIX14T, CSL-daily) are publicly available benchmark datasets commonly used in machine learning research and do not contain personally identifiable information. We have carefully reviewed the ICLR Code of Ethics and confirm that this submission complies with its principles regarding fairness, privacy, and research integrity. No potential conflicts of interest exist among the authors.

## REPRODUCIBILITY STATEMENT

To ensure reproducibility, we provide the following resources: (1) All implementation details, including network architectures, hyperparameters, and training protocols, are described in Section 5.1 and the Appendix. (2) Random seeds are fixed, all results are averaged over multiple runs.

## LLM USAGE STATEMENT

Large Language Models (LLMs) were used in this work solely as a general-purpose writing assistance tool-for example, to improve grammar, clarify phrasing, or check technical terminology in the manuscript. LLMs did not contribute to the conception of the research idea. theoretical analysis, experimental design, or interpretation of results. All scientific content, including equations algorithms, and claims, was developed and verified by the authors. No LLM was used to generate novel technical content or to draft substantial portions of the paper. As required by ICLR policy, we confirm that LLMs are not listed as authors, and we take full responsibility for all content under our names.

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

# A APPENDIX

## A.1 MORE TECHNICAL DETAILS

**Text Encoder & Poes Modalities Decoder Details**   Since our Pose Modalities Decoder contains three parallel decoders, for the sake of convenience, we take the Text-to-Pose branch as an example for detailed introduction. As shown in Figure 7, this branch mainly includes two core parts: the text encoder and the pose decoder.

In the text encoder part, we adopted a stacked structure, which consists of $n$ encoder blocks with exactly the same structure. Each encoder block contains three parts: a Multi-Head Attention layer ($MHA$), two Normalization Layers ($NL$) and a Feed-Forward Layer ($FL$). The calculation process of each block can be expressed as:

$$\tilde{t}_n = NL(FL(MHA(NL(t'_n)) + t'_{n-1})), \tag{12}$$

where $t'_{n-1}$ is the text feature of the previous moment.

Correspondingly, in the Pose decoder part, we also use $n$ decoder blocks with the same structure. Specially, each block contains two MHA layers (the first one is masked MHA, which is used to mask future posture information. Its calculation process can be formalized as:

$$\tilde{p}_{m+1} = PoseDecoder(p'_{1:m}, \tilde{t}_{1:N})$$
$$\Leftrightarrow \begin{cases} z_m = FL(MHA_1(p'_{1:m}) + \tilde{p}_m), m \in [1, M]; \\ \tilde{p}_{m+1} = FL(MHA_2(z_m, \tilde{t}_{1:N})), \end{cases} \tag{13}$$

where $z_m$ is the result after the first self-attention layer $MHA_1$. Here, $MHA_1$ is a self-attention layer with an extra masking operation (*i.e.*, $Q=K=V=p'_{1:m}$) and $MHA_2$ tackles the semantic interaction between sign text and pose sequences (*i.e.*, $Q=z_m$ and $K=V=\tilde{t}_{1:N}$).

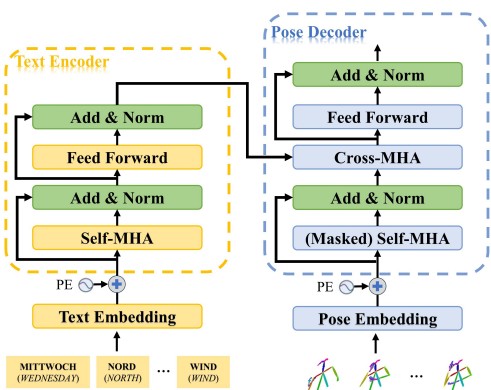

Figure 7: The details of Text Encoder and Pose Decoder.

**RealisticDance Details**   Realisdance is a video generation model based on a dual UNet structure, which is jointly constructed by Reference UNet and Main UNet, as shown in Figure 8. Reference UNet extracts static features of the input image through VAE (Variational Autoencoder) and DINO (Deep Image Feature Extractor) modules, and uses the Self-Attention and Cross-Attention mechanisms in UNet for feature enhancement. Main UNet focuses on dynamic information processing, introduces motion data through the Gate module, and combines the motion perception module with the attention mechanism to model spatiotemporal dependencies. Finally, the two-stage network jointly outputs a video frame sequence, which is optimized by VAE post-processing, significantly improving the visual quality and spatiotemporal consistency of the generated video. This architecture achieves high-quality and natural video generation effects through the decoupling and fusion of static and dynamic features.

## A.2 MORE EXPERIMENTAL DETAILS

Section A.2.1 introduces the hyperparameters used in Text-driven Pose Modalities Co-generation. Section A.2.2 analyzes the semantic and visual effects of different pose modalities. Section A.2.3

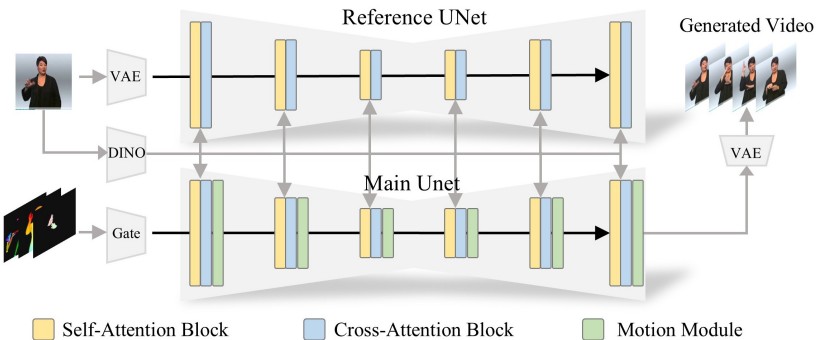

Figure 8: The details of RealisDance.

further analyzes the generation effect of SignAligner with more detailed visual examples. Section A.2.5 describes in detail the composition and implementation methods of different back-translators. Section A.2.4 describes the details of different modality extraction models. Section A.2.6 describes the details of evaluation metrics.

### A.2.1 HYPER-PARAMETERS OF BASELINES

Table 9 presents the hyper-parameters of text-driven pose modalities co-generation used in this work.

Table 9: Hyper-parameters of text-driven pose modalities co-generation.

| Parameters | TextEncoder | PoseDecoder | HamerDecoder | SmplerxDecoder |
|---|---|---|---|---|
| layers | 2 | 2 | 2 | 2 |
| attention heads | 4 | 4 | 4 | 4 |
| hidden size | 512 | 512 | 512 | 512 |
| learning rate | $1 \times 10^{-3}$ | $1 \times 10^{-3}$ | $1 \times 10^{-3}$ | $1 \times 10^{-3}$ |
| optimizer | Adam | Adam | Adam | Adam |
| dropout | 0 | 0 | 0 | 0 |
| batch-size | 64 | 64 | 64 | 64 |
| trg-size | – | 120 | 156 | 96 |

### A.2.2 SEMANTIC AND VISUAL QUALITY ANALYSIS OF DIFFERENT POSE MODALITIES

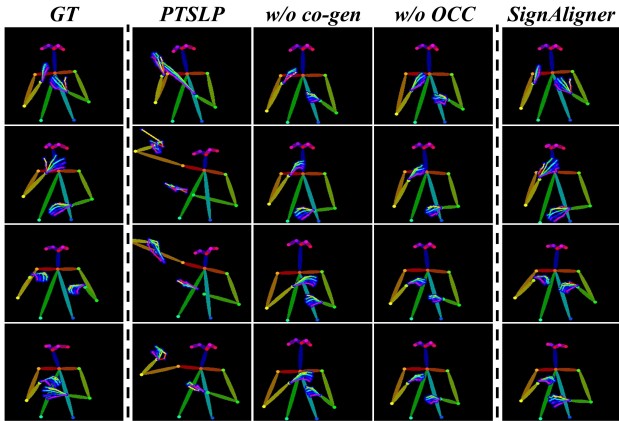

Figure 9: Visualization examples on Text to Pose task.

Table 10: All results on PHOENIX14T for Text to Pose task.

| Methods | DEV | | | | | | TEST | | | | | |
|---|---|---|---|---|---|---|---|---|---|---|---|---|
| | B1↑ | B2↑ | B3↑ | B4↑ | ROUGE↑ | WER↓ | B1↑ | B2↑ | B3↑ | B4↑ | ROUGE↑ | WER↓ |
| PTSLP Saunders et al. (2020) | 12.51 | 6.50 | 4.76 | 3.88 | 11.87 | 96.85 | 13.35 | 7.29 | 5.33 | 4.31 | 13.17 | 96.50 |
| NAT-AT Huang et al. (2021) | – | – | – | – | – | – | 14.26 | 9.93 | 7.11 | 5.53 | 18.72 | 88.15 |
| NAT-EA Huang et al. (2021) | – | – | – | – | – | – | 15.12 | 10.45 | 7.99 | 6.66 | 19.43 | 82.01 |
| DET Viegas et al. (2023) | 17.25 | 10.17 | 7.04 | 5.32 | 17.85 | – | 17.18 | 10.39 | 7.39 | 5.76 | 17.64 | – |
| G2P-DDM Xie et al. (2024b) | – | – | – | – | – | – | 16.11 | 11.37 | 9.22 | 7.50 | – | 77.26 |
| GCDM Tang et al. (2025b) | 22.88 | 14.28 | 10.01 | 7.64 | 23.35 | 82.81 | 22.03 | 14.21 | 10.16 | 7.91 | 23.20 | 81.94 |
| GEN-OBT Tang et al. (2022) | 24.92 | 15.72 | 11.20 | 8.68 | 25.21 | 82.36 | 23.08 | 14.91 | 10.84 | 8.01 | 23.49 | 81.78 |
| w/o co-gen | 19.26 | 11.33 | 7.79 | 5.84 | 20.17 | 87.86 | 18.41 | 11.18 | 7.75 | 5.89 | 19.79 | 87.13 |
| w/o OCC | 19.68 | 11.81 | 8.48 | 6.66 | 19.78 | 89.78 | 19.61 | 11.96 | 8.53 | 6.65 | 19.81 | 89.48 |
| SignAligner (Ours) | **25.13** | **15.88** | **11.87** | **8.71** | **25.33** | **75.55** | **24.39** | **15.61** | **11.09** | **8.47** | **25.21** | **73.89** |

**Analysis of Text to Pose task.** Due to space limitations, Table 4 in the main text only shows some core indicators on the test set that can intuitively reflect the semantic accuracy. To supplement the complete evaluation, we list all evaluation indicators on the validation set and test set in Table 10. It is worth emphasizing that our method outperforms existing mainstream methods in all indicators, especially in terms of WER, which has achieved a significant decrease, further verifying our advantages in semantic modeling and action consistency. In addition, as shown in Figure 9, we compare the baseline method PTSLP, the pose results without Online Collaborative Correction(OCC), and the results generated by our SignAligner. It can be observed in multiple examples that PTSLP produces obviously wrong or distorted pose sequences, while SignAligner can effectively restore the key points of the action and visually present more natural and coherent sign language movements. Its generated results are closer to the ground truth in both morphology and semantics, reflecting the comprehensive advantages of our method in pose accuracy and visual fidelity.

Table 11: All results on PHOENIX14T for Text to Hamer task.

| Methods | DEV | | | | | | | | TEST | | | | | | | |
|---|---|---|---|---|---|---|---|---|---|---|---|---|---|---|---|---|
| | B1↑ | B2↑ | B3↑ | B4↑ | ROUGE↑ | SSIM↑ | PSNR↑ | FID↓ | B1↑ | B2↑ | B3↑ | B4↑ | ROUGE↑ | SSIM↑ | PSNR↑ | FID↓ |
| PTSLP Saunders et al. (2020) | 13.75 | 7.84 | 5.58 | 4.42 | 13.67 | 0.948 | 19.041 | 24.007 | 13.26 | 7.71 | 5.45 | 4.32 | 13.03 | 0.948 | 19.061 | 25.652 |
| GEN-OBT Tang et al. (2022) | 22.75 | 14.99 | 11.35 | 9.15 | 23.48 | 0.951 | 19.848 | 19.626 | 22.44 | 14.66 | 10.76 | 8.52 | 21.87 | 0.951 | 19.769 | 19.999 |
| CogvideoX Yang et al. (2024) | 15.42 | 8.79 | 6.26 | 4.89 | 14.67 | 0.915 | 14.924 | 36.093 | 15.16 | 9.14 | 6.83 | 5.62 | 14.17 | 0.915 | 14.932 | 36.587 |
| LVMCN Wang et al. (2025) | 21.95 | 14.45 | 10.76 | 8.56 | 22.25 | 0.952 | 19.986 | 21.953 | 22.20 | 14.61 | 10.74 | 8.48 | 22.23 | 0.951 | 19.903 | 22.475 |
| w/o co-gen | 24.40 | 16.07 | 12.03 | 9.56 | 25.08 | 0.953 | 20.227 | 6.292 | 24.06 | 15.97 | 11.93 | 9.55 | 23.73 | 0.952 | 20.184 | 6.302 |
| w/o OCC | 27.61 | 18.70 | 14.05 | 11.16 | 27.79 | 0.958 | 21.124 | 5.541 | 27.98 | 19.15 | 14.33 | 11.43 | 27.59 | **0.958** | 21.107 | 5.552 |
| SignAligner (Ours) | **30.09** | **20.52** | **15.36** | **12.19** | **29.89** | **0.959** | **21.319** | **4.510** | **29.94** | **20.74** | **15.53** | **12.27** | **29.12** | 0.958 | **21.314** | **4.428** |

**Analysis of Text to Hamer task.** To further analyze the performance of Text-to-Hamer, we supplemented the complete semantic evaluation results in Appendix Table 11 that are not shown in Table 5 of the main text due to space limitations. Overall, SignAligner achieved the best performance in all semantic indicators on the validation set and test set, with BLEU-4 and ROUGE scores of 12.27% and 29.12% on the test set, respectively, significantly outperforming the existing methods PTSLP (B4: 5.42%, ROUGE: 13.03%) and CogvideoX (B4: 6.52%, ROUGE: 14.17%). This fully demonstrates that our method has significant advantages in generating semantically consistent and well-structured sign language representations. In addition, compared with the ablation experiments that removed the pose modalities co-generation (**co-gen**) or Online Collaborative Correction (**OCC**), the complete SignAligner model has further improved in all indicators, verifying the effectiveness of the (**co-gen**) and (**OCC**). In terms of image quality, SignAligner also achieved the best results in terms of SSIM (0.958), PSNR (21.314) and FID (4.428), further demonstrating its strong capabilities in visual fidelity and spatiotemporal consistency. In particular, the FID index was reduced by 21.224 and 32.159 compared with PTSLP (25.652) and CogvideoX (36.587), respectively, indicating that the generated video is closer to the real data in distribution and the visual effect is more natural and credible.

Moreover, we show the Hamer of various methods in Figure 10. It can be observed that PTSLP and CogvideoX have gesture blur, unclear structure or motion misalignment in multiple frames, especially in some high-dynamic actions, the generated results are far behind Ground Truth. Although the results of co-generation (w/o OCC) have greatly solved the problem of motion misalignment, there are still problems in finger details. In comparison, the gesture morphology generated by Sig-

nAligner is clearer and more accurate, and is overall closer to the true label than the result before correction, verifying the effectiveness of our proposed SignAligner in modeling semantic and temporal information.

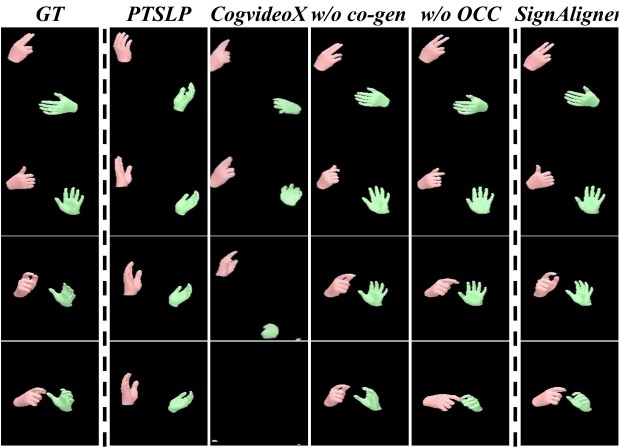

GT  PTSLP  CogvideoX  w/o co-gen  w/o OCC  SignAligner

Figure 10: Visualization examples on Text to Hamer task.

**Analysis of Text to Smplerx task.**   Table 12 shows the fully quantified results of Text to Smplerx on the PHOENIX14T dataset. Compared with the existing methods PTSLP and CogvideoX, SignAligner has also achieved significant advantages in all evaluation indicators, verifying the comprehensive improvement of our method in semantic expression and visual quality. On the test set, SignAligner achieved 27.48%, 18.78%, 14.00%, and 11.20% in BLEU-1 to BLEU-4, and 27.43% in ROUGE, which are much higher than PTSLP (BLEU-4 is only 3.36%) and CogvideoX (BLEU-4 is 2.45%), indicating that the generated action sequence is more accurate in semantic consistency with the reference sequence. In addition, in the image quality evaluation, SignAligner's SSIM is 0.832, PSNR is 18.651, and FID is reduced to 3.607, which is significantly better than PTSLP and CogvideoX, showing higher visual fidelity and spatiotemporal consistency. At the same time, the ablation experiment results also show that the pose modalities co-generation (co-gen) and online collaborative correction (OCC) modules play a key role in performance improvement. Removing any module will cause a significant decrease in indicators such as BLEU, ROUGE and FID, further proving the effectiveness of SignAligner's design and the synergy between modules.

Table 12: All results on PHOENIX14T for Text to Smplerx task.

| Methods | DEV | | | | | | | | TEST | | | | | | | |
|---|---|---|---|---|---|---|---|---|---|---|---|---|---|---|---|---|
| | B1↑ | B2↑ | B3↑ | B4↑ | ROUGE↑ | SSIM↑ | PSNR↑ | FID↓ | B1↑ | B2↑ | B3↑ | B4↑ | ROUGE↑ | SSIM↑ | PSNR↑ | FID↓ |
| PTSLP Saunders et al. (2020) | 10.35 | 4.66 | 2.92 | 2.19 | 10.25 | 0.790 | 16.104 | 7.415 | 9.89 | 4.85 | 3.36 | 2.62 | 9.65 | 0.790 | 16.228 | 7.584 |
| GEN-OBT Tang et al. (2022) | 24.56 | 16.52 | 12.37 | 9.91 | 26.06 | 0.803 | 16.777 | 3.848 | 25.87 | 17.52 | 13.16 | 10.55 | 25.43 | 0.803 | 16.844 | 3.968 |
| CogvideoX Yang et al. (2024) | 13.31 | 5.74 | 2.69 | 1.50 | 11.05 | 0.712 | 12.047 | 62.301 | 9.85 | 4.46 | 2.45 | 1.56 | 7.67 | 0.724 | 12.033 | 43.228 |
| LVMCN Wang et al. (2025) | 22.79 | 15.32 | 11.52 | 9.27 | 24.19 | 0.806 | 16.985 | 4.596 | 23.75 | 16.11 | 12.04 | 9.77 | 24.25 | 0.807 | 17.047 | 4.106 |
| w/o co-gen | 18.84 | 12.42 | 9.49 | 7.75 | 21.10 | 0.806 | 17.003 | 6.882 | 19.89 | 12.87 | 9.88 | 8.01 | 19.98 | 0.808 | 17.068 | 6.842 |
| w/o OCC | 19.97 | 12.87 | 9.69 | 7.91 | 21.33 | 0.832 | 18.560 | 4.879 | 20.38 | 13.49 | 10.24 | 8.33 | 20.87 | 0.832 | 18.582 | 4.861 |
| SignAligner (Ours) | **25.42** | **16.98** | **12.66** | **10.04** | **26.00** | **0.831** | **18.619** | **3.570** | **27.48** | **18.78** | **14.00** | **11.20** | **27.43** | **0.832** | **18.651** | **3.607** |

Figure  11 shows a visual comparison of Smplerx sequences generated by different methods.  It can be observed that the sequences generated by PTSLP and CogvideoX have obvious defects, such as incoherent gestures, blurred motion, distorted posture structures, etc., and overall show poor temporal consistency and motion expression capabilities. In contrast, SignAligner maintains a clear and stable posture trajectory in each frame, with natural and smooth movement transitions and good structural consistency. Its generated sequence is closer to Ground Truth in terms of spatial configuration and dynamic evolution trend. Especially at key action nodes, SignAligner shows higher control accuracy and is significantly better than other methods in terms of arm coordination and movement range consistency. This result not only verifies its sophisticated ability to model postures under semantic drive, but also reflects its structural control advantage in complex dynamic action generation.

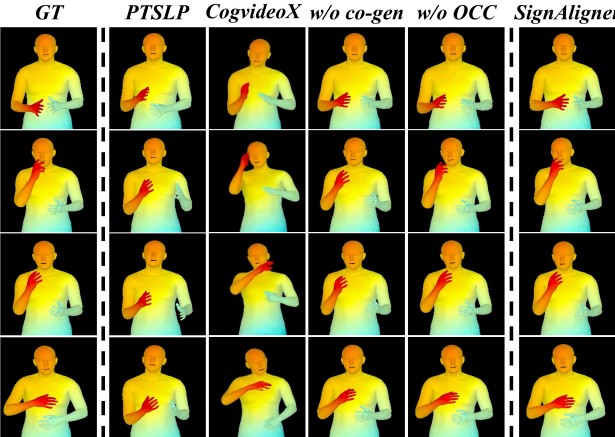

Figure 11: Visualization examples on Text to Smplerx task.

### A.2.3 VISUALIZATION EXAMPLES OF SIGNALIGNER

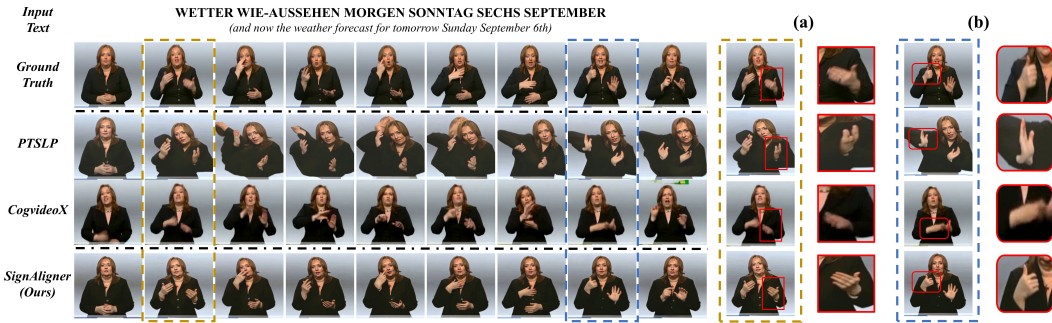

Figure 12: Visualization examples of produced sign language video sequence on PHOENIX14T dataset. We compare our method with PTSLP Saunders et al. (2020), CogvideoX Yang et al. (2024) and the ground truth.

Figure 12 shows the visual comparison results of sign language video sequences generated by different methods on the PHOENIX14T dataset. We selected an input text "WETTER WIE-AUSSEHEN MORGEN SONNTAG SECHS SEPTEMBER" as an example and compared the video frames generated by Ground Truth, PTSLP, CogvideoX and our proposed SignAligner. Overall observation shows that SignAligner can generate more natural, coherent and close to real video sequences, with clear action boundaries, natural expressions, and strong spatial consistency and temporal coherence.

The yellow dotted box (a) in the figure shows the frame area with incorrect labels. We found that although Ground Truth has a certain degree of annotation deviation, SignAligner can still robustly generate frames with reasonable structure and correct actions. In contrast, the outputs of PTSLP and CogvideoX are more susceptible to incorrect labels, and the actions are offset or missing. The blue dotted box (b) marks the keyframes of action details. From the enlarged red box area, it can be observed that SignAligner shows higher fidelity and visual detail restoration capabilities in gesture edges, action amplitudes, and hand postures. For example, PTSLP has blurred or misaligned hands in some frames, while CogvideoX's action connection is not natural enough and easily causes frame skipping. The above analysis shows that SignAligner not only shows stronger robustness in the face of real label noise, but also has better modeling capabilities in action details and temporal structures. The generated results are closer to real videos, verifying its comprehensive advantages in the sign language video generation task.

### A.2.4 Details of data extraction from different modalities

**Pose.** We use the efficient full-body state estimation model DWPose Yang et al. (2023b) to extract the 2D coordinates of 60 key points across the human body, including joints, hands, and face, with poses $\in R^{T \times 60 \times 2}$. Unlike the traditional OpenPose Cao et al. (2017) method, DWPose achieves high-precision keypoint extraction through an innovative two-stage refinement strategy combined with hierarchical knowledge transfer from a teacher-student model. It demonstrates higher accuracy in hand details, especially when processing large movements.

**Hamer.** In sign language video analysis, hand movements serve as the core semantic carriers, and their refined spatial representations are crucial for accurately conveying semantic information, especially the subtle deformations of the finger joints. Therefore, we adopt the state-of-the-art 3D gesture estimation method HaMeR Pavlakos et al. (2024), which is based on a transformer model trained on a large-scale hand dataset, and is able to provide high-quality 3D and depth information about the hand for more accurate understanding of the spatial structure of gestures. Specifically, we decode two sets of key parameters from the pre-trained model: the hand gesture parameter $\theta_{\mathrm{h}} \in R^{16 \times 3}$ (characterizing the joint rotations and displacements) and the shape parameter $\beta_{\mathrm{h}} \in R^{10}$ (describing the morphological differences of individual hands). These parameters are mapped by the function $M(\theta_{\mathrm{h}}, \beta_{\mathrm{h}})$ to a high-resolution hand mesh $M \in R^{V \times 3}$, where $V = 778$ vertices are connected by triangular facets to form a complete 3D hand model.

**Smplerx.** We estimate expressive human pose and shape parameters by means of a 3D parametric human model, SMPLer-X Cai et al. (2023). Specifically, we first estimate the pose parameters $\theta_{\mathrm{s}} \in R^{55 \times 3}$, including body, hand, eye, and chin poses; joint body, hand, and face shapes $\beta_{\mathrm{s}} \in R^{10}$ as well as facial expressions $\psi \in R^{10}$ by pretraining the model, and then model the body, hand, and face geometries by combining the parameters through a joint regressor, which results in 10475 3D mesh vertices, and finally use rendering techniques to draw the Video. In this way, the rendered results simultaneously integrate 3D spatial, depth and continuous semantic information, thus effectively compensating for the lack of conventional skeletal data that contains only 2D information. However, compared with Smplerx, Hamer has higher accuracy in complex gesture estimation, so we use Hamer to supplement the hand information in Smplerx.

### A.2.5 Construction of different back-translators

**Back translation model NSLT.** Based on previous research Saunders et al. (2020); Huang et al. (2021); Tang et al. (2022), we used the back-translation model NSLT Camgoz et al. (2018) as an evaluation tool for our Text to Pose task. To achieve fairness in the experiment, we retrained a fair back-translator NSLT-new on the same setting using our data PHOENIX14T. We can obtain BLEU, ROUGE, WER and other indicators that can describe the semantic accuracy of pose in detail. NSLT-new is only used for the evaluation of subsequent experiments.

**Back translation model GFSLT.** In order to comprehensively evaluate our different sign language representations - Hamer, Smplerx and Video, we adopted the current advanced sign language translation method GFSLT Zhou et al. (2023) as a unified evaluation framework, which can obtain multiple semantic accuracy indicators including BLEU, ROUGE, etc. Specifically, we used Hamer, Smplerx and video in the dataset PHOENIX14T as training data, and independently trained the corresponding evaluation models according to the training strategy given by GFSLT. This method has a strong ability to restore text semantics from sign language expression, and can complete high-quality semantic decoding. Therefore, with the help of GFSLT, we can directly evaluate the effectiveness of different sign language representations in semantic communication. Through a unified training process and evaluation criteria, the fairness and scientificity of the comparison are ensured, and the quality of the content generated by SignAligner in terms of semantic expression and visual restoration can also be verified.

### A.2.6 More Details of Evaluation Metrics

In the experimental section, we use multiple metrics for quantitative analysis from different dimensions. We use **SSIM**, **PSNR** and **FID** to measure the visual similarity between the generated videos and the real videos at the image and video levels. In addition, we use **BLEU**, **ROUGE** and **WER**

Table 13: Evaluation indicator classification.

| Dimensions | Metrics |
|---|---|
| Visual similarity | SSIM, PSNR, FID |
| Semantic quality | BLEU, ROUGE, WER |

to evaluate semantic performance. Subsequently, we will interpret the above evaluation metrics in more detail.

**SSIM (Structural Similarity Index):** The SSIM Wang et al. (2004) is a perceptual metric that quantifies image quality degradation by comparing structural information, luminance, and contrast between a reference and a distorted image. It ranges from -1 to 1, where 1 indicates perfect similarity.

- $x$: reference image patch
- $y$: distorted image patch
- $\mu_x, \mu_y$: mean intensities of $x$ and $y$
- $\sigma_x^2, \sigma_y^2$: variances of $x$ and $y$
- $\sigma_{xy}$: covariance between $x$ and $y$
- $C_1 = (K_1 L)^2, C_2 = (K_2 L)^2$: stabilization constants, where $L$ is the dynamic range (255 for 8-bit images), $K_1 = 0.01$, $K_2 = 0.03$
- The SSIM formula is:

$$\text{SSIM}(x, y) = \frac{(2\mu_x\mu_y + C_1)(2\sigma_{xy} + C_2)}{(\mu_x^2 + \mu_y^2 + C_1)(\sigma_x^2 + \sigma_y^2 + C_2)} \tag{14}$$

- For full images, Mean SSIM (MSSIM) is calculated by averaging SSIM values over all patches

**PSNR (Peak Signal-to-Noise Ratio):** The PSNR Hore & Ziou (2010) is a widely used metric for measuring the quality of reconstructed images. It quantifies the ratio between the maximum possible power of a signal and the power of corrupting noise, expressed in decibels (dB). Higher PSNR values indicate better image quality.

- $I$: reference image of size $m \times n$
- $K$: reconstructed image of same size
- $\text{MAX}_I$: maximum possible pixel value (255 for 8-bit images)
- MSE: mean squared error between images

$$\text{MSE} = \frac{1}{mn} \sum_{i=0}^{m-1} \sum_{j=0}^{n-1} [I(i, j) - K(i, j)]^2 \tag{15}$$

- The PSNR formula is:

$$\text{PSNR} = 10 \cdot \log_{10}\left(\frac{\text{MAX}_I^2}{\text{MSE}}\right) \tag{16}$$

**FID (Fréchet Inception Distance):** The FID Heusel et al. (2017) measures the similarity between generated and real image distributions using features from the Inception network. Lower FID scores indicate better quality and diversity of generated images (perfect match = 0).

- $p_r$: real image distribution
- $p_g$: generated image distribution
- $\mu_r, \mu_g$: mean features of real and generated images from Inception-v3 pool3 layer
- $\Sigma_r, \Sigma_g$: covariance matrices of real and generated features
- The FID formula is:

$$\text{FID} = \|\mu_r - \mu_g\|^2 + \text{Tr}\left(\Sigma_r + \Sigma_g - 2\left(\Sigma_r\Sigma_g\right)^{1/2}\right) \tag{17}$$

where Tr denotes the matrix trace operation.

**BLEU (Bilingual Evaluation Understudy):**   The BLUE Papineni et al. (2002) is a commonly used metric to assess the quality of machine translation. It is calculated by the following formula:

$$\text{BLEU} = \text{BP} \cdot \exp\left(\sum_{n=1}^{N} w_n \cdot \log p_n\right) \tag{18}$$

where:

- BP (Brevity Penalty): A brevity penalty factor used to penalize candidate translations that are shorter than the reference translations.

$$\text{BP} = \begin{cases} 1 & \text{if } c > r \\ e^{(1-\frac{r}{c})} & \text{if } c \leq r \end{cases} \tag{19}$$

- $c$ denotes the length of the candidate translation.
- $r$ denotes the length of the reference translation.
- $p_n$: The precision for n-grams, defined as the number of matching n-grams between the candidate and reference translations divided by the total number of n-grams in the candidate translation.

$$p_n = \frac{\text{Number of matched n-grams}}{\text{Total number of n-grams in candidate translation}} \tag{20}$$

- $w_n$: The weight assigned to each n-gram precision, typically set as $w_n = \frac{1}{N}$, where $N$ represents the maximum length of the n-grams considered (commonly $N = 4$, covering 1-gram through 4-gram precision).
- $exp$: Represents the exponentiation of the sum of the weighted logarithmic precisions.

In our main experiments, we mainly use BLEU-1 and BLEU-4 scores to reflect the accuracy of word-level translation and the quality of overall sentence translation, respectively.

**ROUGE (Recall-Oriented Understudy for Gisting Evaluation):**   The ROUGE Lin (2004) is a metric based on the Longest Common Subsequence (LCS), used to evaluate the sequence and content matching between generated text and reference text. It captures the similarity in sentence structure. We set it to $1.2$.

- $X$ be the reference sentence with a length of $m$, and $Y$ be the generated sentence with a length of $n$. $\beta$ represents the ratio of precision to recall.
- $LCS$: The longest subsequence of elements that appear in both sequences in the same order.
- $R_{CSL}$: Indicates the ratio of the LCS length to the length of the reference text.

$$R_{CSL} = \frac{LCS(X,Y)}{m} \tag{21}$$

- $P_{CSL}$: Indicates the ratio of the LCS length to the length of the generated text.

$$P_{CSL} = \frac{LCS(X,Y)}{n} \tag{22}$$

- $F_{LCS}$: The harmonic mean of LCS recall and precision.

$$F_{LCS} = \frac{(1+\beta^2)R_{LCS}P_{LCS}}{R_{LCS} + \beta^2 P_{LCS}} \tag{23}$$

**WER (Word Error Rate):**   The WER **?** is a commonly used evaluation metric to measure the accuracy of a translation system. WER measures the error rate in the generated text, accounting for three types of errors: substitutions, insertions, and deletions. It is expressed as the ratio of the total number of errors to the total number of words in the reference text. The formula for calculating WER is:

$$\text{WER} = \frac{S + D + I}{N} \tag{24}$$

where:

- $S$ represents the number of substitutions.
- $D$ represents the number of deletions.
- $I$ represents the number of insertions.
- $N$ is the total number of words in the reference text.

