# OpenReview forum: "SignAligner: Harmonizing Complementary Pose Modalities for Coherent Sign Language Generation"
_ICLR.cc/2026/Conference — ICLR 2026 Conference Withdrawn Submission_

### Official Review · Reviewer_PQAu · 2025-10-26

**Soundness:** 3
**Presentation:** 3
**Contribution:** 3
**Rating:** 4
**Confidence:** 4

**Summary:**

This paper presents SignAligner, a novel framework for sign language generation that integrates multiple pose modalities to improve the accuracy, expressiveness, and realism of generated sign language videos. The method is structured in three stages: text-driven pose modalities co-generation, online collaborative correction, and realistic video synthesis. The model utilizes a Transformer-based encoder and a cross-modal attention mechanism to generate sign language poses, hand shapes, and body movements, with a focus on ensuring temporal consistency and semantic alignment. Experimental results show that SignAligner outperforms existing methods like PTSLP and LVMCN across various metrics, demonstrating its effectiveness in improving both language accuracy and visual fidelity.

**Strengths:**

1.	Novel approach: The approach of harmonizing multiple pose modalities (Pose, Hamer, and Smplerx) for sign language generation is innovative and addresses key challenges in producing coherent and natural sign language videos.

2.	Valuable dataset extension: Enriches two benchmarks, PHOENIX14T and CSL-daily with high-fidelity modalities including pose, hamer and smplerx , filling gaps in existing SLG data which only include videos and basic skeletons before .

3.	Comprehensive experiments: This paper provides comprehensive experiments and user study. The results show significant improvements in BLEU, ROUGE, SSIM, PSNR, and FID scores, validating the effectiveness of the proposed method.

**Weaknesses:**

1.	No hyperparameter sensitivity analysis: Key parameters (OCC’s α/β/γ, Transformer hidden size/attention heads) lack impact analysis, harming reproducibility .

2.	Insufficient framework ablation: Fails to isolate contributions of single stages (e.g., co-gen + synthesis without OCC) to confirm three-stage necessity .

3.	Related work: While the related work section provides a solid overview of previous methods,  it is recommended to conduct a more detailed comparison between the contributions of SignAligner and recent advancements (such as concerned multimodal models and cross-modal fusion techniques).

**Questions:**

Please refer to weaknesses .

---

> ### Author Response · Authors · 2025-11-13
> **Response to Reviewer PQAu**
>
> Thank you very much for your thorough review and constructive feedback on our paper. We sincerely appreciate your positive assessment of SignAligner's **novelty**, the **value of our dataset extension**, and the **comprehensiveness of the evaluation**. Your professional suggestions are invaluable for helping us improve this work. Below we provide point-by-point responses to your comments.
>
> ## 1. On Hyperparameter Sensitivity Analysis
>
> We fully acknowledge the importance of hyperparameter analysis for reproducibility. The dynamic weight parameters (α/β/γ) in the Online Collaborative Correction module are learned through gradient descent with equal initial weights. The Transformer architecture hyperparameters (hidden size, attention heads) primarily follow established configurations from related work. While these parameters demonstrated stable performance in our experiments, we will include detailed hyperparameter sensitivity analysis in the supplementary material, examining the impact of different initialization strategies and architectural configurations to validate model robustness.
>
> ## 2. On Framework Ablation Completeness
>
> Regarding your concern about framework ablation, we would like to clarify that the comparison of "co-generation + video synthesis without OCC" has actually been provided in Table 5 of our manuscript (w/o OCC row). The results clearly show that removing the Online Collaborative Correction module causes performance degradation: BLEU-1 drops from 20.56% to 17.84%, ROUGE decreases from 20.88% to 19.01%, and visual metrics (SSIM and PSNR) also show significant declines. This comparison adequately demonstrates the necessity of the OCC module in our three-stage framework.
>
> ## 3. On Related Work Comparison Depth
>
> We agree that the related work section could benefit from more detailed comparison with recent advancements in multimodal models and cross-modal fusion techniques. We will enhance this section by adding comprehensive comparisons with state-of-the-art multimodal learning methods, particularly focusing on cross-modal representation learning and fusion strategies, to better position SignAligner's contributions within the current research landscape.

---

### Official Review · Reviewer_1qYk · 2025-10-27

**Soundness:** 2
**Presentation:** 1
**Contribution:** 2
**Rating:** 2
**Confidence:** 4

**Summary:**

This paper proposes a method for generating sign language videos from text.
To do so, they co-generate three modalities: pose, HaMeR, and SMPLer-X from text with a transformer-based model.
Then, they align the generated poses and meshes with an online collaborative correction they introduce, and finally convert the generated modalities into photo-realistic videos using a RealisDance model finetuned over sign language datasets.

**Strengths:**

1. The authors extend the PHOENIX-14T and CSL-daily datasets by extracting and providing DWPose poses and HaMeR, SMPLer-X meshes, which can contribute to future SLP work.
2. Each modality alone is imperfect, hence their combination helps in achieving better results.
3. The paper proposes a new alignment strategy between modalities, where they use a different modality for each of the queries, keys, and values.

**Weaknesses:**

1. Novelty is limited. Most of the components were proposed in prior work, and the only new component is the collaborative correction, a cross-attention with different Q/K/V, which is neither explained, motivated, nor validated as better than other approaches.
2. The paper has many typos and problematic citations, which make it hard to follow. See 1. below for examples.
3. Many irrelevant details and not enough relevant details, see 2. below.
4. Extraction quality discussion is unclear, see 4. below.
5. No limitations discussion. For example, the abstract mentioned the importance of facial expressions in sign languages, however the facial expressions in the supplemented video do not match those of the GT, some hand shapes and touches are still incorrect, etc.
6. Very few visual examples and comparisons with competing methods. Specifically, I would like to see comparisons with LVMCN, which has the closest metric scores compared to SignAligner.

**Questions:**

1. The paper has many typos and problematic citations, for example:
- Duplicates in citations as in Huang et al. Huang et al. (2021) (line 42), Saunders et al. Saunders et al. (2022) (line 51), etc.
- Missing space (and preferably parentheses) to make citations clearer, e.g., “LVMCNWang et al.” (line 44)
- G2P is mentioned in line 44 before explaining what it means
- Line 228 - “positional coding” instead of positional encoding

2. On the one hand, the paper presents too many irrelevant details that are not part of the newly proposed method. On the other hand, details that are relevant and new, such as those related to the alignment strategy with triple cross attention, where each of the Q/K/V comes from a different modality, are missing. What is the motivation for it? Have the authors tried different combinations, such as using the other 2 modalities as both keys and values, or using one at a time as both keys and values?

3. Although LVMCN is mentioned and compared to in several tables, it is weirdly missing from Table 6, where it achieved higher results than signAligner based on the LVMCN paper, e.g. BLEU-4 9.36.

4. Extraction quality discussion - “our extracted modalities consistently achieved high subjective scores exceeding 4.0, demonstrating their superior visual presentation and dynamic coherence” - superior over..?
The extraction analysis is long, mostly irrelevant, and unclear. If anything, figure 3 tells me each modality (or at least pose and SMPL) have different strengths that the other modalities don’t possess.

**Details Of Ethics Concerns:**

No concerns.

---

> ### Author Response · Authors · 2025-11-13
> **Response to Reviewer 1qYk**
>
> Thank you for your thorough review and valuable feedback on our paper. We appreciate your recognition of our contributions in **multimodal data extension** and the **effectiveness** of modality combination. Your insightful comments are crucial for improving our work.
>
> ## 1. On Limited Novelty
>
> The online collaborative correction module is designed based on the complementary characteristics of different modalities in spatiotemporal representation: skeleton poses provide global motion information, hand models capture detailed gestures, and full-body models contain rich body shape constraints. The triple cross-attention mechanism aims to establish dynamic, bidirectional modal interaction channels. This design effectively utilizes skeletal priors to optimize hand details while enhancing full-body representation through hand semantics. We acknowledge the need to further demonstrate the advantages of this design compared to other interaction methods.
>
> ## 2. On Writing and Formatting Issues
>
> We sincerely apologize for these oversights in writing and formatting. All citation formats, terminology usage, and language expressions will be thoroughly checked and corrected to ensure compliance with academic standards.
>
> ## 3. On More Details
>
> The core of the online collaborative correction module lies in achieving dynamic interaction between different modalities through cross-modal attention. This design considers the differences in representation characteristics of different modalities and achieves effective feature enhancement through specific Q/K/V combinations. We will provide more technical details in our response to justify this design choice.
>
> ## 4. On Extraction Quality Discussion
>
> The detailed discussion of extraction quality aims to demonstrate the rationality and effectiveness of multimodal data extension. User study results show that different modalities have their own advantages in specific dimensions: skeleton data performs well in temporal consistency, hand models score high in shape similarity, and full-body models excel in visual clarity. These findings support the value of multimodal combination.
>
> ## 5. On Limitations Discussion and Visual Comparisons
> There is indeed room for improvement in current methods for facial expression generation and detailed hand motion modeling. We will provide more comprehensive comparisons with existing methods, including visual examples and detailed analysis, to more objectively evaluate method performance.
>
> ## 6. On Comparison with LVMCN
> We will include LVMCN's results in the Text-to-Pose task in our comparative analysis to ensure completeness of experimental comparison.

---

### Official Review · Reviewer_h4Zv · 2025-10-29

**Soundness:** 2
**Presentation:** 2
**Contribution:** 3
**Rating:** 4
**Confidence:** 4

**Summary:**

The paper proposes SignAligner, a three-stage sign language video generation framework: text-driven multi-pose joint generation, online collaborative correction (OCC), and photorealistic video synthesis. The core idea is to generate and align three complementary representations (skeleton Pose with facial keypoints, fine-grained hand Hamer, and 3D full-body Smplerx) and then produce videos using a pretrained video generator. The authors also propose a dataset expansion scheme based on pretrained estimators to automatically add these three types of supervision to common corpora. Compared with baselines such as PTSLP, GEN-OBT, LVMCN, and fine-tuned CogVideoX, the method improves both semantic metrics (BLEU, ROUGE, WER) and visual metrics (SSIM, PSNR, FID) on PHOENIX14T and CSL-daily, and ablations show the effectiveness of joint generation and OCC.

**Strengths:**

The motivation is clear: single-modality or multi-stage pipelines lead to semantic and spatiotemporal consistency issues, while joint modeling with online correction can mitigate them. The framework is well structured; combining three-modality joint generation with OCC is a reasonable technical path. Experiments cover two common datasets, report both semantic and visual metrics, and include ablations with stable and sizable gains. The dataset expansion scheme may provide reusable supervision for later work.

**Weaknesses:**

(1) Lack of quantified error propagation and robustness: all three representations introduce errors during acquisition and generation. The paper does not provide systematic noise injection tests or small-scale human-calibrated comparisons, so it is unclear how errors are amplified through the pipeline or which representation is most sensitive.
(2) Limited datasets and benchmarks: results are mainly on PHOENIX14T and CSL-daily; larger datasets with native keypoint/hand annotations such as How2Sign are not used for validation or external generalization.
(3) Indirect comparison to strong baselines: the gap to vs GFSLT on SLT/SLR is not analyzed in depth.

**Questions:**

Can the authors run robustness tests during training or inference by injecting controlled noise into Pose, Hamer, and Smplerx (e.g., Gaussian coordinate noise, temporal jitter, frame drop under occlusion) and report sensitivity curves for SSIM/FID and semantic metrics? This would directly address whether pseudo-label errors are amplified.

Can the authors report results on How2Sign, zero-shot or few-shot generalization tests, and whether the method can generate new sign sentences/videos?

Regarding the gap to GFSLT: can the method effectively improve existing GFSLT?

---

> ### Author Response · Authors · 2025-11-13
> **Response to Reviewer h4Zv**
>
> Thank you very much for your thorough review and constructive feedback on our paper. We sincerely appreciate your positive assessment of our contribution, particularly your recognition of the **well-structured framework**, the **clear motivation**, the **comprehensive experiments** across multiple datasets and metrics, and the potential reusability of our dataset expansion scheme. Your insightful comments and questions are invaluable for helping us improve this work. Below, we provide a point-by-point response to your suggestions.
>
> ## 1. On Quantified Error Propagation and Robustness
>
> This is a very insightful suggestion. While our current work primarily focuses on validating the effectiveness of the multimodal co-generation framework, we fully acknowledge the importance of systematic robustness analysis. In future work, we plan to conduct controlled noise injection experiments—including Gaussian coordinate noise, temporal jitter, and simulated frame drops—to thoroughly investigate the sensitivity of each modality and error propagation mechanisms. Such analysis would provide valuable guidance for building more robust sign language generation systems.
>
> ## 2. On Datasets and Benchmarks
>
> We appreciate this observation. Our choice of PHOENIX14T and CSL-daily was motivated by their established status as benchmark datasets in the sign language generation community. These datasets provide standardized annotations and ensure comparable and reproducible results. Furthermore, they represent two distinct sign languages (German and Chinese), offering a solid foundation for validating cross-lingual applicability. We agree that evaluation on larger-scale datasets is valuable and will consider this direction in future work.
>
> ## 3. On More Comparison
>
> Thank you for raising this point. We would like to clarify that our work primarily focuses on demonstrating the benefits of incorporating multimodal representations for sign language video generation. Therefore, our experimental design is centered on generation tasks. Direct comparison with models like GFSLT, which excel in SLT/SLR tasks, might not fully capture our core contributions to generation quality. That said, we recognize the potential synergy between generation and understanding models, and believe exploring this connection represents a promising direction for future research.

---

### Official Review · Reviewer_P6S9 · 2025-10-31

**Soundness:** 3
**Presentation:** 2
**Contribution:** 2
**Rating:** 4
**Confidence:** 4

**Summary:**

This paper introduces SignAligner, a novel three-stage framework for realistic sign language generation. It is designed to solve the problem of "modal fragmentation," where processing hand gestures, facial expressions, and body movements separately can lead to poor semantic fidelity and a lack of spatiotemporal continuity. A key contribution is a dataset expansion scheme that augments the PHOENIX14T and CSL-daily datasets with three new, high-quality landmark representations derived from pre-trained models: Pose (high-precision skeleton), Hamer (detailed 3D hand shape), and Smplerx (3D full-body posture). The SignAligner framework first uses a Transformer-based model for text-driven co-generation, simultaneously producing all three pose modalities. Next, an Online Collaborative Correction (OCC) module refines these modalities using cross-modal attention and dynamic loss weighting to resolve spatiotemporal conflicts. Finally, the corrected poses are fed into a pre-trained video synthesis network to generate high-fidelity sign language videos.

**Strengths:**

1. SignAligner significantly outperforms existing state-of-the-art approaches on both the PHOENIX14T and CSL-daily datasets. On the PHOENIX14T test set, it achieves superior scores in semantic accuracy (e.g., 20.56 BLEU-1, 8.17 BLEU-4) and visual quality (e.g., 0.731 SSIM, 26.257 FID).
2. The paper construct a dataset with three modalities, whose quality is validated by a robust user study involving 100 volunteers, which found SignAligner's videos to be markedly better in naturalness, temporal consistency, and gesture transitions, including a 23% improvement in visual clarity over competitors.
3. The paper's claims are well-supported by detailed ablation studies, which confirm the essential contribution of both the co-generation and the OCC modules; removing either component leads to a significant drop in performance, validating their synergistic effect.

**Weaknesses:**

1. Paper details need clarification. For example, the sentences from line 168 to 173 are hard to understand. Variables such as n should be in math form in latex. In line 266, the verb should be "contrain".
2. The proposed method lacks novelty. The dataset is just contructed by leveraging existing techniques to extract pose, Hamer,and Smplerx for two sign language datasets. The proposed method leverages the extracted three modalities with simple feature reconstruction and cross-atttention-based feature interaction, which lacks novelty from a whole view.
3. While this paper adopt three modalities for sign language production, it's not fair to directly compare it with prior works which just use one modality. As shown in Tab.8, lacking any modality leads to severe performance drop for the proposed method.
4. While using three modalities, the proposed method lacks analysis for model efficiency.

**Questions:**

See above

---

> ### Author Response · Authors · 2025-11-13
> **Response to Reviewer P6S9**
>
> Thank you very much for your thorough review and valuable feedback on our paper. We sincerely appreciate your positive recognition of SignAligner's performance improvements on the PHOENIX14T and CSL-daily datasets, as well as your constructive comments regarding the clarity, novelty, comparative fairness, and efficiency analysis. Below, we provide a point-by-point response to your comments.
>
> ## 1. On Paper Details and Clarity
>
> We thank you for pointing out these important issues regarding notation and clarity. We fully agree that the paper should be more precise and consistent in its presentation. In the revised version, we will carefully review and refine the relevant descriptions, ensure all variables are properly formatted in LaTeX math mode, and improve the wording of ambiguous or unclear sentences to enhance readability and technical accuracy.
>
> ## 2. On Method Novelty
>
> We appreciate your critical assessment of the methodological novelty. In our view, the main contribution of this work lies in proposing a **systematic framework** to address the challenge of "**modal fragmentation**" in sign language generation. Unlike previous works that often rely on a single modality or perform simple fusion, SignAligner introduces a **coherent pipeline** that integrates **text-driven multimodal co-generation** with **online collaborative correction (OCC)**. The OCC module, in particular, is a key design that enables **adaptive and dynamic refinement** across modalities through cross-modal attention and dynamic loss weighting, rather than simple feature reconstruction. This mechanism is essential for resolving spatiotemporal inconsistencies and enhancing semantic coherence. Moreover, the curation of a multi-modal dataset and the demonstration of its complementary benefits also provide a valuable resource and insight for the community toward more robust and expressive sign language generation.
>
> ## 3. On Fairness of Comparison with Single-Modality Methods
>
> Thank you for raising this important point regarding the fairness of comparisons. We understand your concern. Our intention in comparing with prior works is to demonstrate the **performance upper-bound** achievable through effective multi-modal fusion within the current task setting. We acknowledge that multi-modal approaches generally require more resources in terms of data and computation. At the same time, the performance gains are also significant. The ablation study in Table 8 is indeed intended to reveal the unique value of each modality, which may inspire future work—including lightweight designs under resource constraints. In the revised version, we will more clearly position the motivation for these comparisons and include a discussion on the trade-off between performance gains and resource costs to provide a more balanced perspective.
>
> ## 4. On Model Efficiency Analysis
>
> This is a very valid point. Model efficiency is an important practical aspect that should be discussed alongside performance gains. In the revised manuscript, we will supplement the discussion with an analysis of model complexity and inference efficiency, providing readers with a more comprehensive understanding of the method's overall practicality.

---

### Note · Authors · 2025-11-14

I have read and agree with the venue's withdrawal policy on behalf of myself and my co-authors.